# Studying Stickiness: Methods, Trade-Offs, and Perspectives in Measuring Reversible Biological Adhesion and Friction

**DOI:** 10.3390/biomimetics7030134

**Published:** 2022-09-15

**Authors:** Luc M. van den Boogaart, Julian K. A. Langowski, Guillermo J. Amador

**Affiliations:** 1Experimental Zoology Group, Department of Animal Sciences, Wageningen University & Research, De Elst 1, 6708 WD Wageningen, The Netherlands; 2Department of BioMechanical Engineering, Faculty of Mechanical, Maritime and Materials Engineering, Delft University of Technology, Mekelweg 2, 2628 CD Delft, The Netherlands

**Keywords:** biological adhesion, friction, contact mechanics, biomimetics, force sensor, bioinspiration

## Abstract

Controlled, reversible attachment is widely spread throughout the animal kingdom: from ticks to tree frogs, whose weights span from 2 mg to 200 g, and from geckos to mosquitoes, who stick under vastly different situations, such as quickly climbing trees and stealthily landing on human hosts. A fascinating and complex interplay of adhesive and frictional forces forms the foundation of attachment of these highly diverse systems to various substrates. In this review, we present an overview of the techniques used to quantify the adhesion and friction of terrestrial animals, with the aim of informing future studies on the fundamentals of bioadhesion, and motivating the development and adoption of new or alternative measurement techniques. We classify existing methods with respect to the forces they measure, including magnitude and source, i.e., generated by the whole body, single limbs, or by sub-structures. Additionally, we compare their versatility, specifically what parameters can be measured, controlled, and varied. This approach reveals critical trade-offs of bioadhesion measurement techniques. Beyond stimulating future studies on evolutionary and physicochemical aspects of bioadhesion, understanding the fundamentals of biological attachment is key to the development of biomimetic technologies, from soft robotic grippers to gentle surgical tools.

## 1. Introduction

Controlled reversible attachment is a key adaptation across diverse terrestrial animal groups that exhibit various locomotory modes and encounter complex three-dimensional environments. Sticking to vertical or overhanging substrates requires a combination of strong adhesion (i.e., attachment force perpendicular to a substrate) and strong friction (i.e., attachment force parallel to a substrate) [1]. Among spiders, insects, tree frogs, and geckos, various versatile attachment strategies have evolved. The adhesive pads on the limbs of geckos and spiders rely on what is commonly referred to as ‘dry’ adhesion, thought to be dominated by weak intermolecular forces [2,3,4], while those of insects and tree frogs are believed to rely also on what is referred to as ‘wet’ adhesion—liquid-mediated interactions, such as capillary and viscous forces [5,6,7]. In addition to the adhesive pads on their limbs, animals may utilize other body parts to control or aid their attachment, such as generating friction through other tarsal segments in insects [8] or through the belly in tree frogs [9,10], or using claws to mechanically interlock with asperities on substrates [11,12]. These mechanisms have been studied in animals that vary in size across several orders of magnitude—from insects and spiders of a couple of milligrams to geckos and tree frogs of several hundreds of grams in mass [13].

Some animals can rapidly establish and reverse attachment, with stride frequencies of up to 10 steps per second for geckos or even 100 steps per second for mites [8]. To achieve such rapid reversibility, animals presumably control the strength of their attachment via shear-sensitive adhesive pads and control peeling by varying the angle between their limb and the substrate [8,14,15]. Furthermore, there is increasing evidence that shearing and peeling also contribute to self-cleaning during locomotion [16,17,18,19].

The fundamental understanding of rapid and reversible attachment of biological systems can inform many biomimetic applications that benefit humans in daily life. Reversible adhesion finds applications in sticky tapes, robotic grippers [20,21,22], and climbing robots [23,24,25]. The development of surgical tools may be inspired by the strategies and mechanisms used by animals, specifically for the manipulation of delicate and slippery tissues inside the human body [26,27,28]. Other applications can be found in agriculture and architecture, such as the development of grippers for autonomous harvesting robots [29], protecting crops from animal pests [30,31], improving pollination of flowers [32,33], protecting buildings from termites [34], or safeguarding people from disease vectors such as mosquitoes and ticks [35,36,37].

Accurate measurements of adhesion and friction forces are crucial for unravelling the fundamental mechanisms of biological adhesion, or bioadhesion. In order to understand and transfer the underlying principles of bioadhesion into biomimetic applications, physicochemical models of attachment need to be developed and validated against experimentally measured attachment forces, or derived parameters such as normal or shear stresses. As adhesive forces correlate strongly with contact area [38], normalizing adhesion forces to average adhesive stresses using contact areas provides a scale-independent representation of adhesive capacity [3]. However, measuring these parameters accurately poses a number of challenges.

To measure maximum adhesion and friction performance, one needs to detach the animal from a substrate through external forcing. These external forces can be applied globally, as a field, like gravitational or centrifugal forces, or locally by pulling on parts of the animal, for example through a tether. Such forces can be applied to the entire animal, one of its organs, or its sub-structures. In force measurements of live animals, behavior needs to be considered. When an animal moves freely it might employ behavioral strategies that are different than when it is perturbed, constrained, or sedated. Isolating individual limbs (or sub-structures) can help to control for animal behavior; however, extrapolating measurements on a single limb to the whole animal may lead to errors due to assumptions and oversimplifications. For example, in some animals, it has been found on the limb-level that larger adhesive pads generate stronger adhesion per unit area [13,39,40,41], which, however, may be explained through behavioral adaptations on the whole-organism-level (i.e., active shearing of the pad for adhesion control; [41]).

Given the many parameters that can influence adhesion and friction, such as temperature, humidity, and substrate properties, as well as the hierarchy of biological attachment devices (Figure 1), many factors need to be considered in the design of a bioadhesion study. In this review, we give an overview of the methods used for measuring contact forces in animal attachment studies, and discuss their trade-offs. This review limits itself to methods used in studies on terrestrial animals because they have direct implications for applications that humans encounter in their daily, (mostly) terrestrial life. However, many of the methods presented here are also used in studies on aquatic bioadhesive systems.

We conclude this review with a novel perspective on force measurement methods focusing on force magnitudes and how they are generated by and/or applied to the animal. To this extent, we will review the most-used force measurement methods considering whole animals (Figure 1A), isolated limbs (Figure 1B), and their sub-structures (Figure 1C), and whether the animals experience global or local forcing. Additionally, we address relevant parameters that can be measured, controlled, and varied in the different methods. This overview provides guidance for scientists that are new to the field of bioadhesion, and presents key challenges in measurement methodology that need to be overcome to advance the field. To assist those new to the field, we also provide a glossary of technical terms at the end.

## 2. Force Measurement Methods

In the past four decades, numerous measurements methods have been used in bioadhesion research. Table 1 outlines these methods, including the animals they have been used on, variables they measure and control for, and the ranges of force magnitudes they are capable of measuring. In the ensuing text of this section, we describe the methods in detail and elaborate on how they have been implemented in previous bioadhesion studies.

### 2.1. Global External Forcing

**Force platforms** (Figure 2A) are the most commonly used method to measure the contact forces of climbing animals. Conventional three-dimensional (3D) force platforms allow for the measurement of the magnitude and direction of ground reaction forces during locomotion and attachment. These measurements can be used to characterize gait patterns of studied animals, determining attachment forces through calculating stabilizing moments during locomotion [42]. The simplest setups consist of a single force platform for recording reaction forces [42,44].

The main limitation of using a single force platform for the entire animal is the inability to distinguish force contributions from individual legs. To compensate for this limitation, later studies present experimental setups with an increased spatial resolution by using multiple platforms in force measurement arrays (FMAs). FMAs have been predominantly used to investigate the gait patterns of lizards [43] and tree frogs [15,47,48]. Reinhardt et al. [45] and Endlein & Federle [46] used custom-built force platforms with *μ*N-resolution to measure the reaction force of a single leg of an ant during climbing. While 3D force platforms are among the few methods that allow simultaneous measurement of frictional and adhesive forces, it is typically impossible to measure the contact area during attachment due to constraints of the setup design space. Increasing the spatial resolution of the force platform to enable contact stress measurements would require multiple individual sensors for each adhesive pad, which quickly becomes impractical due to growing costs and time needed for calibration and data analysis.

**Optic tactile sensors** (Figure 2B) have been developed to measure contact areas and forces of adhesive pads during locomotion. By enabling the visualization of contact area, this method addresses one of the major limitations of force platforms. Such sensors exploit optical phenomena, like light refraction, to highlight areas where an adhesive organ comes into contact with a substrate, and measure substrate deformations to quantify contact forces. Earlier optic contact area sensors used in insect studies worked with photo-elastic gelatin [49], making use of polarizing filters to measure substrate deformation and, as a result, ground reaction forces. This method, however, is limited in substrate selection.

Later optic sensors exploit frustrated total internal reflection (FTIR), a technique first developed by Betts et al. in 1980 [83]. FTIR works by trapping a beam of light inside of a transparent substrate of high refractive index compared to air, e.g., glass, which has a refractive index of 1.5 compared to 1.0 for air. By shining light into the substrate at a shallow angle, the light will reflect internally and, when an object comes into contact with the substrate, the relative reflective index will be lowered locally, allowing light to escape and highlight the contact area. FTIR is limited by camera resolution. Stride frequencies of up to 100 strides per second require adhesion to be established and reversed within milliseconds [8]. Therefore, capturing the dynamics of such events requires a high temporal resolution, which in cameras typically conflicts with the high spatial resolution that is needed to record adhesion events in small animals like insects (e.g., the leg of a mosquito has a diameter of 50 µm [36]). Having both high temporal and spatial resolutions requires efficient data processing procedures and cameras with high quantum efficiency sensors. This makes tactile optic sensors and FTIR good alternatives for slow and large animals.

Eason et al. [52] developed an advanced FTIR-based sensor to measure the adhesive stress distribution of a gecko foot during climbing. This sensor makes use of a polymeric sensing membrane covered in flexible pyramidal bumps, named taxels, placed atop an acrylic waveguide. When force is applied to the membrane, the taxels buckle and the contact area between the sensing membrane and the waveguide increases, causing more light to scatter. This way measurable light intensity is related to the applied pressure, allowing the mapping of stress distributions during contact at high spatial and temporal resolutions (about 60 taxels per mm^2^ at 60 Hz). FTIR has mostly been used for tree frogs in a completely free animal experiment [51] and in combination with rotation platforms [10,15]. Federle & Endlein [50] have also successfully used FTIR to image contact area in ants, measuring areas of several hundreds of *μ*m^2^ at frame rates of up to 250 Hz.

**Rotation platforms** (Figure 2C) provide a way to vary the orientation of an animal relative to the gravitational field. After the animal is placed on a horizontal platform, the platform is rotated around a horizontal axis until the animal is pulled off by gravity. The angle of the platform at which the animal drops off can be used to quantify adhesive force, with a completely inverted platform coinciding with an adhesive force equal to (or greater than) the animal’s weight. For this reason, rotation platforms are limited to animals whose safety factor (SF: the ratio of attachment force to body weight) is one or lower. Rotation platforms have been used to study tree frogs [9,10,15,18,55], salamanders [84], beetles [53], mirid bugs [54], and ticks [37]. This technique can be relatively easily combined with FTIR to measure contact areas and determine average stresses. Moreover, the rotation platform is minimally invasive (i.e., animals are unconstrained) and the substrate can be easily exchanged or modified (e.g., covered with a liquid film; [9]). Because of the typically high SF of insects and arachnids, rotation platforms are not well-suited for adhesion force measurements in these animals, but can instead be used to compare the probability of attachment to different substrates [37]. While rotation platforms are ideally suited for studies with animals with SF of one or lower, there are no explicit upper or lower limits for the magnitudes of forces they can measure. In Table 1, the lower bound for the measurable force range coincides with previous measurements on mirid bugs [54].

**Adhesion force centrifuges** (Figure 2D.i) are the most frequently used alternative to rotation platforms for insects. When used to measure adhesive forces, the studied animal is placed on the side of a drum or vertical platform attached to a horizontal arm. The drum or arm is then rotated around a vertical axis at increasing angular velocities (typically up to 3000 revolutions per minute) until the centrifugal force exceeds the adhesive force and the studied animal detaches. This method is effective for animals with high SFs. Force centrifuges are able to record forces in a range between around {500 µN, 500 mN} (lower bound: motor precision, upper bound: maximum motor rotation speed). These ranges could be expanded by optimizing the centrifuge motor. Centrifuging techniques used to measure adhesion forces were first introduced by Dixon et al. [85] and later used to study ants [40,56,57,58], moths [59], and stick insects [41].

The effectiveness of force centrifuges is limited by subject mass. Since centrifugal forces are directly proportional to subject mass, special care needs to be placed on the structural robustness of the setup when scaling up. Moreover, the high centrifugal forces required to overcome high adhesion result in greater impact forces after release, which increases the risk of injury for test subjects. This makes using force centrifuges for heavier animals ethically challenging. Centrifuges are best suited for insect studies, or for animals with masses in the range of {1 mg, 1 g}. Gorb et al. [61] concluded that the influence of aerodynamic drag on force measurements using centrifuges is negligible in insects; however, aerodynamic forces may become significant for larger or non-streamlined specimens.

**Friction force centrifuges** (Figure 2D.ii) use the principle of controlling centrifugal forces for measuring static and dynamic friction forces, similar to adhesion force centrifuges. In friction force centrifuge measurements, subjects are placed on top of a horizontal disk or drum that rotates around a vertical axis. A laser or camera is used to monitor the subjects’ distance from the center of the disk. Measuring the tangential acceleration and the centrifugal force component, the friction force can be calculated. Keeping the rotational velocity constant after static friction is overcome by the centrifugal force, dynamic friction can be calculated by tracking the sliding displacement and deriving acceleration. Like adhesion force centrifuges, friction force centrifuges are most effective for insects with high SFs and low body mass. Most friction force centrifuge experiments are based on a setup developed by Gorb et al. [61]. This or a similar setup has been used to study ants [86], beetles [62,63], coddling moths [64], sawfly larvae [60], and syrphid flies [61].

### 2.2. Local Forcing

#### Whole Animal Measurements

**Adhesion force tethers** (Figure 3A.i) provide a simple way to quantify adhesive forces. They are cheap and easy to set up, but invasive, as they require a strain gauge or scale to be attached to the animal, which may trigger unnatural postures or unwanted reactions due to induced stress. For example, a tethered study on the Tokay gecko *Gekko gecko* was carried out by Pugno et al. [71]. While the study clearly showed a decreasing trend in adhesive force over multiple trials due to foot damage, it underestimated the adhesive capacity of the gecko by more than a factor of 30. The authors suggested this to be the result of ‘imperfections’ on the toes; however, another likely explanation lies in the forced posture. The subject’s limbs were pulled to unnatural angles wherein it was unable to fully engage its adhesive structures. While it is challenging to prevent such effects, synchronised video recordings of the animal may help monitor for induced changes in posture and for effects of tether location on measured attachment forces.

**Friction force tethers** (Figure 3A.ii) are the most commonly used method to measure friction forces. Like in adhesion force tether experiments, the studied animal is attached with a wire to a strain or force gauge. Alternatively, it is possible to pull on an animal positioned on a force sensor. The method was first used by Walker et al. [66] in a study on blowflies, in which the substrate was pulled while the animal remained stationary. This way, dynamic friction was measured for various pulling directions. Later studies measured static friction by making the animal walk over the substrate, pulling on the force transducer. Reviewed studies suggest a measurable force range of {200 µN, 10 mN} due to sensor limitations. All studies reviewed made use of the same force transducer: 10 g capacity, Biopac Systems Ltd., Santa Barbara, CA, USA. Bounds could be expanded by using force transducers of a higher capacity or sensitivity.

The method has been used as means of validation for force centrifuge tests [57], as well as to study the attachment of insects to various substrates. Examples include studying the effects of free surface energy [67], substrate roughness [68], or substrate chemistry [69] on insect friction. Additionally, tethered animal trials have been used to study the attachment capacity of insects to various plants [30,31,70] and flower petals [32].

### 2.3. Limbs and Below

In fundamental studies into the physicochemical basis of biological attachment, the behavior of the animal should be controlled for, such as when investigating how sub-structures enable adhesion and friction, or how adhesion and friction together contribute to attachment. In these cases, it makes sense to isolate the body part or sub-structure of interest. Doing so increases the controllability of the experiment and enables the measurement of forces in greater detail than in whole animal studies. Moreover, when we exclude confounding factors due to animal behavior from the experiment, we can more accurately estimate the maximum capacity of an adhesive system.

**Force transducers** (FTs; Figure 3B) are widely used to measure adhesion and friction of the toes, pads, and sub-structures (like fibres, setae, or spatulae) of geckos, frogs, and insects. Different configurations have been developed in various studies, but most of them are either uniaxial or biaxial FTs. Uniaxial FTs in limb studies are mostly used to measure adhesive forces. Biaxial FTs, mounted to a translation stage in combination with a closed loop controller, can be used to keep adhesive forces constant to isolate frictional forces or measure adhesive forces while applying shear loads.

Several types of sensors have been used. Uniaxial FTs typically rely on fibre optic springs [65,72] or piezoelectric sensors [19]. Spinner et al. [73] used an uniaxial FT to measure friction forces, by sliding the feet of a chameleon over a rod attached to the FT. Biaxial FTs mostly rely on strain gauges placed in perpendicular directions [2,3,18,74,75,76]. Force transducers are able to record forces in a range between {80 µN,100 mN} (lower bound: sensor precision [41], upper bound: sensor limitation of the 10g force transducers used). One study by Autumn et al. [77] used a 3-axis force sensor to measure the friction force of an array of setae from a gecko for various loading directions. One study by Gillies et al. [78], also on a gecko, used a 6-axis force sensor, though this was presumably due to availability. Keeping the amount of measuring axes to a minimum is beneficial since it reduces the amount of calibration needed, controller complexity, financial costs, and data processing.

**Atomic force microscopy** (AFM; Figure 3C,D) is an indispensable method in bioadhesion research, either to measure adhesive or frictional forces directly or functioning in a supportive role. AFM relies on the optical or piezoresistive sensing of the deflection of a cantilever, which is brought into contact with a substrate. AFM can measure adhesion forces with a resolution of 70 pN [79,80,81]. This makes AFM suitable to measure adhesive forces in a range of around {200 pN,1 µN} (lower bound: roughly three times the precision of 70 pN [79], upper bound: maximum force in flexible probe range [87]). AFM is not limited to a specific animal group or animal weight because it measures at a very small spatial range, e.g., at the (sub-)setal range. AFM has, for example, been used to measure the adhesive capacity of gecko setae [3,79], capillary forces on the terminal plates of fly setae [82], the friction profile across individual substrate features on the toe pads of tree frogs [88], and the adhesion of a mosquito limb on rough substrates [36].

## 3. Discussion

In the previous section, we presented a broad overview of existing methods to study the attachment of terrestrial animals (see Table 1 for a summary). When deciding on a method for a new study, one should consider a few questions. What parameters need to be measured (e.g., force, contact area, stress)? What are the magnitudes of the parameters to be measured? Is the method suitable for the animal of interest? Does the method provide the freedom to choose and/or vary experimental conditions (e.g., substrate characteristics)? Does the method limit the behavior of the animal? Are there alternative methods available for the study?

In this section we present relevant considerations when selecting a method. First, we consider some of the limitations of the most prevalent methods with respect to scale and subject, e.g., species and body part. Then, the trade-offs in selecting a method for a study are discussed. Lastly, we present outlooks for future development and the general implications of animal adhesion studies in science and society.

### 3.1. Limitations

When deciding on a method, it is critical to consider the size of the animal and magnitudes of the attachment forces it can generate. Figure 4 shows a regime map of the most common adhesion and friction force measurement methods. Only AFM, 2D (biaxial) FTs, tethers, rotation platforms, and force centrifuges are included. Force platform studies are excluded because they include both whole animal and limb measurements, as well as 1D, 2D, and 3D force measurements, and so are difficult to compare. To our knowledge, there are insufficient previous studies (n<3) available in the literature to make meaningful estimates of the regimes of photo-elastic gelatin and optic tactile sensors. However, their limitations were discussed in the previous section.

In Figure 4, measured force is plotted against subject mass as reported in the reviewed studies. The data shows two distinct trends: (1) whole animal studies follow constant safety factor (SF) lines, and (2) body part measurements are limited by sensor precision. The measurable ranges are also outlined in Table 1 and in more detail in the foregoing section.

As noted before, rotation platform limits are explained well by the animal’s SF, which should be considered during the design of an experiment. For the other whole animal force measurement methods, tethers and force centrifuge measurements, SF bounds are suggested as well by the reviewed studies. Tethered studies are not effective when SF <1, since animals that can not sustain their own weight through friction will likely start slipping when pulling their own body weight. There is a considerable overlap between tethers and force centrifuge studies, suggesting both are capable of studying the same animal species, and expected SF or animal weight does not need to be considered when choosing between the two. However, Federle et al. [57] report higher adhesion forces for ants when measured using a centrifuge compared to a tether. They speculate that tethers (i.e., local forcing) may affect an animal’s posture and natural response more than a centrifuge (i.e., global forcing).

Considering body part measurements, there is a clear gap between AFM and FTs. The bounds of these methods are set by sensor limitations. Reviewed studies and sensor limitations suggest a gap in the {1 µN, 10 µN} range, above the maximum flexible probe range of AFM and below the sensor precision of FTs. When forces in this range are expected, extra consideration should be taken in designing the measurement setup. Notably, both methods are suitable for any type of animal and are not limited by animal weight because these methods are used to investigate limbs or their sub-structures.

### 3.2. Trade-Offs in Study Design

In addition to animal size and expected force magnitudes, there are other factors to consider when deciding on a method to measure bioadhesion. First, one needs to determine if measurements should be carried out on whole animals or their limbs or sub-structures. Measurements with whole animals are influenced by behavior (e.g., motivation) and body kinematics (e.g., posture). However, investigating behavior may shed light on the postures and kinematics that animals use to promote attachment. For example, observations on tree frogs found that when attaching to overhanging substrates they spread their limbs away from their body to presumably minimize the angle between their limbs and substrate to prevent peeling [15].

While some behaviors promote attachment, there are others that may hinder it. Bioadhesion measurements only work when animals attach to substrates and do not jump or fly away. Insects capable of flight may need to be incapacitated by gluing or trimming their wings to prevent escape. In their study with moths, Al Bitar et al. [59] had to cut the insects’ wings to prevent them from fleeing during measurements using force centrifuges. Such modifications allow measuring of attachment forces, but may affect the animal’s behavior and response to external stimuli.

For fundamental studies into the physicochemical basis of attachment, bioadhesion measurements are best carried out with individual limbs or their sub-structures, where animal behavior can be controlled for. These measurements enable control over kinematics and mechanics, and thus may provide a deeper insight into the mechanisms underlying the generation of adhesion and friction. For example, previous work using individual limbs has found that the adhesive pads of geckos, tree frogs, and insects are shear-sensitive and generate increased adhesion under enhanced shear loading [8]. The linear relationship between shear force and adhesive force would be impossible to observe with whole animals. By working with individual limbs and biaxial FTs, the shear forces were controlled while adhesive forces measured.

In another example, the adhesive forces generated by a single gecko seta were carefully measured using AFM [2]. Then, the measured forces were compared with predictions from an analytical model of van der Waals forces (i.e., the interaction forces between the molecules on the seta and the substrate) to test if such intermolecular forces underpin gecko adhesion [3]. This finding motivated the development of gecko-inspired, micro-structured adhesives that stick without glue by also exploiting van der Waals forces [89]. Therefore, bioadhesion studies using limbs or their sub-structures have the potential to generate fundamental knowledge of great importance for the design of biomimetic adhesives.

As stated in the introduction, in order to measure attachment performance, an animal needs to experience an external force that works against the adhesion and friction it can generate. This external force can be applied globally, as a field, or locally, and the way it is applied can significantly influence the study. Global forcing is typically done using gravitational or centrifugal forces. These force fields act on the whole animal uniformly and simulate the forcing that an animal may experience when attaching to vertical or overhanging substrates. Local forcing acts on individual body parts. While such forcing is not typically experienced by animals in day-to-day life, it enables the isolation of individual limbs (and their sub-structures) and provides minimalist ways to measure maximum attachment performance, e.g., tethered studies require only a thin wire and force sensor.

Finally, the parameters that need to be measured and controlled, i.e., the dependent and independent variables, respectively, should be identified. Table 1 outlines the dependent and independent variables that were measured and controlled in previous studies. Based on this, tethers, force transducers, and AFM are the most versatile methods. They enable variation and control of independent variables, especially substrate properties and interaction kinematics as well as mechanics. Force platforms and optical methods are the most limited with respect to independent variables. This is primarily because the substrates cannot be controlled or varied due to requirements dictated by the methods, e.g., force platforms have sensors embedded and optical methods require substrate transparency.

### 3.3. Beyond Adhesion and Friction Measurements

While this review focuses primarily on techniques used for measuring forces, there are other parameters that need to be measured to fully grasp the attachment of a given animal. Theoretical models of contact mechanics and attachment can help identify underlying physicochemical mechanisms, but require validation through comparisons with experimental observations. Typically, the models predict adhesion and friction forces that can be compared to measured values; however, the models also depend on additional parameters as inputs.

One particularly important parameter needed in theoretical models of contact mechanics is the distance between the adhesive pad (or its sub-structures) and substrate. The magnitude of this distance could help determine which types of interactions are dominant or negligible. For example, for 10-*μ*m spherical particles under dry conditions, electrostatic forces from the net charge on the particles dominate for distances greater than 100 nm, electrostatic forces from local charge patches dominate for distances between 10 and 100 nm, and van der Waals forces dominate for distances less than 10 nm [90]. Furthermore, if there is fluid present, measuring fluid film thickness can help determine if the fluid acts like a lubricant or enhances friction.

These distances can be measured through interference reflection microscopy (IRM). This technique was first developed to measure how close cells are to substrates [91], but was later used with tree frog toe pads [76,92]. In tree frogs, it was found that while mucus is present on the toe pads, parts of the surface features on frog toes are in quasi-direct contact with the substrate, with separation distances between 0 and 35 nm [76], indicating a potential contribution of van der Waals forces or other ‘dry’ interactions in tree frog attachment. Additionally, it was found that there is an intermediate fluid film thickness (∼200 nm) that enhances friction compared to a fully wet (lubricating) or fully dry state [92].

Fluids covering the contact surface are an inherent part of many bioadhesive systems. For example, tree frog toes—as the whole amphibian body—are covered with a watery mucus [93], and insects secrete a viscous emulsion onto their adhesive pads. While these fluids help to prevent skin and cuticle from drying out and may have anti-bacterial and anti-fungal properties [94,95], their implications in bioadhesion are still being investigated. The physical and chemical properties of these fluids have been measured using various techniques. To measure the fluid’s viscosity, methods were adopted from the field of rheology. For tree frog mucus, laser optical tweezers were used to measure the viscous force exerted on a trapped particle by the mucus [76]. The viscosity of insect pad fluid was measured by placing small tracer particles in a drop of the fluid and recording the dampening of the particle’s Brownian motion (or thermal fluctuations) through the fluid’s viscosity [96].

For chemical characterisation of the fluid, several techniques have been used. In tree frogs, cryo-histochemistry, attenuated total reflectance-infrared spectroscopy, and sum frequency generation spectroscopy have been used. From the measurements, it was found that the mucus on the toe pads is chemically similar to the mucus secreted by other body parts, including the belly [93]. In insects, gas chromatography and mass spectrometry have been used to characterize the chemical composition of their secreted fluids [97]. From this characterization, it was found that, like in tree frogs, the fluid secretions on the adhesive pads are chemically similar to those secreted throughout the rest of the body [98].

Surface tension is another important physical property of a bioadhesive fluid, as the capillary forces associated with it can be dominant at small spatial scales. However, to our knowledge, this property so far has been measured only indirectly through contact angle measurements [58,99,100]. Contact angle, or the angle between the substrate and fluid meniscus, quantifies the ‘wettability’ of a fluid on a substrate. For insects and tree frogs, this contact angle has been found to be quite small (~10°) on a wide variety of substrates, so the adhesive fluid appears to be highly wetting regardless of substrate chemistry [58,99,100]. Recent studies of insects have made assumptions of the surface tension of the fluid given that it is comprised of hydrocarbons [101,102]. This assumed, approximate value sufficed for these studies since the models provided leading order analyses of the capillary interactions. For more detailed and accurate models, direct characterization will be required.

The material properties of the pad tissues, setae, or spatulae are also important for understanding bioadhesion. Animals stick to a wide variety of substrates, including smooth and rough ones. For rough substrates, the adhesive pads should conform to asperities in order to form a large area of close contact. A pad’s ability to conform to rough substrates is dictated by its physical properties, especially its stiffness or Young’s modulus. This property can be measured using micro- or nano-indentation, where the adhesive pad is compressed by a small probe and its stress response is measured, or using optical techniques, like confocal laser scanning microscopy [103]. Using such techniques, it has been found that setae on the adhesive pads of beetles are stiffer at the base and softer at the tip [103]. Similarly, the smooth adhesive pads of insects exhibit softer tissues in the outer layers and stiffer tissues underneath [104]. On the other hand, for tree frogs, it was found that the outer layers of the toe pads are stiffer than internal tissues [105,106].

Pad stiffness not only influences conformability, but may also affect the strength of adhesion. Classical experiments measuring the adhesion between a spherical indenter and flat substrate found that adhesion increases with material stiffness [107]. Similarly, the attachment force of fiber-reinforced adhesives such as gecko toes is proportional to the tensile stiffness of the fiber-reinforcement [108]. Therefore, there seems to be a trade-off between having soft pad tissues to conform to rough substrates and having stiff tissues to generate strong adhesion. In geckos and tree frogs, blood sacks have been observed immediately underneath the adhesive skin surface. Blood pressure may be controlled in these sacks to help tune pad stiffness [106,109]. Having such control could enable geckos and tree frogs to easily conform to rough or non-flat substrates using soft tissues and then stiffen the tissues to promote strong adhesion. A similar mechanism has been exploited by synthetic adhesives that use phase changing liquid metals [110].

AFM is a very versatile method that allows more than just contact force measurements. Many studies that investigate the effects of substrate properties on attachment use AFM to measure roughness, or to image surface sub-structures. Alternatives for measuring surface roughness of biological samples, such as scanning electron microscopy (SEM), are prone to artefacts from the preparation steps, such as shrinkage or drying, and are not suitable for living animals [105]. AFM can also be used for indentation experiments. Micro-indentation using FTs with a motorized stage is sufficient for larger structures, such as whole tree frog toes [106]. However, for smaller structures, AFM is required, for example to measure the stiffness of epithelial cells and local friction profiles over single pillars on tree frog toes [88], or the stiffness of the adhesive tarsal setae of ladybird beetles [103].

While the physical and chemical properties of adhesive pads and their fluid secretions are important for developing physicochemical models of adhesion and friction, the ways in which contact is established and released, i.e., pad and limb kinematics, can significantly influence attachment and detachment. Previous work has found that animals may be able to control adhesion by varying shear forces [8,65]. In addition to controlling adhesion via shear, tree frogs have been observed to spread out their limbs away from their body in response to increased loads [15]. By spreading their limbs, they not only promote shearing but also decrease the angle between their limbs and substrate. Just like in sticky tapes, minimizing this angle may prevent peeling. For insects, it has been found that attachment and detachment occur at different time scales [111,112]. Specifically, adhesive pads move quicker during detachment, which is believed to help conserve the secreted fluid. A faster separation velocity ensures that less fluid is deposited on the substrate. Additionally, a slower approach during attachment may help generate intimate contact and reduce the gap between pad and substrate to increase adhesion and friction forces [112].

### 3.4. Perspectives

Based on the reviewed data, we could map established force measurement methods to show their effectiveness and limitations, as summarized in Table 1 and Figure 4. From this analysis, we find that studying attachment for the large and slow no longer poses a problem. The frontier lies at the small and fast. Measuring small and fast processes still poses a considerable challenge given the trade-offs in spatial and temporal resolutions for cameras and sensors. There is renewed interest in optical methods during the past decade [9,10,47,51]. With visual data processing technologies, data storage and transfer capacities, and optic systems ever improving, optics-based methods seem promising, like the optic-tactile sensor developed by Eason et al. [52] to directly measure adhesive stress.

Quantifying adhesive and frictional stresses can help reveal the true performance of biological adhesives, since it provides a scale-independent measure of adhesion and friction and captures the exact contact stress distribution. Typically, adhesive pads are asymmetric and limbs are rarely oriented completely parallel or perpendicular to a substrate; therefore, forces are applied with offsets that induce moments and cause imbalances in contact stress distribution. Direct measurements of contact stress distribution can pinpoint where stress concentrations occur to reveal how the adhesive may fail and how limb kinematics influence adhesion and friction. However, to our knowledge, optic-tactile sensors are the only ones capable of contact stress measurements at the moment. Measuring adhesive and frictional stresses across various animals could contribute significantly to our understanding of the scaling of adhesive performance in biological systems [13].

In this review, we have largely skipped over micro-electromechanical sensors (MEMS). Interest in MEMS for measuring attachment seemingly faded in the past decade, but MEMS might be key in exploring the realm of fast and small. A MEMS force plate for studying insect locomotion developed by Bartsch et al. [113,114] has barely been cited in actual animal studies. The same holds for a biaxial MEMS cantilever design by Lin & Tramer [115]. This raises the question: is MEMS irrelevant to bioadhesion research, or have developments in MEMS design gone unnoticed in bioadhesion research?

Bioadhesion has always been a fascinating subject to study for biologists and engineers alike. Their work over the last decades resulted in various insights into these remarkable mechanics, attracting an ever-increasing interest from various other disciplines. Electrical engineers, (soft) roboticists, medical engineers, material scientists, and ecologists all benefit from discoveries in bioadhesion and work to tackle multidisciplinary problems, such as protecting honey bees, preventing animal pests, or developing new soft grippers for various applications.

## Figures and Tables

**Figure 1 biomimetics-07-00134-f001:**
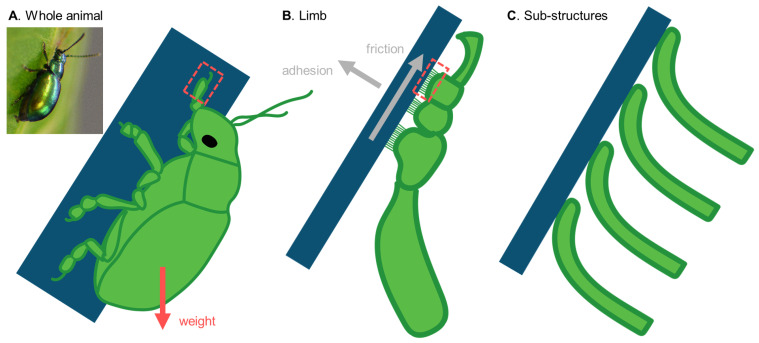
**Levels at which bioadhesion can be studied in an animal.** Schematics of a beetle sticking to a sloped substrate, showing (**A**) the whole animal, (**B**) its limb, and (**C**) sub-structures (fibres, setae, or spatulae, depending on species). Inset of (**A**) shows a green dock beetle *Gastrophysa viridula* on a dock leaf *Rumex* spp.

**Figure 2 biomimetics-07-00134-f002:**
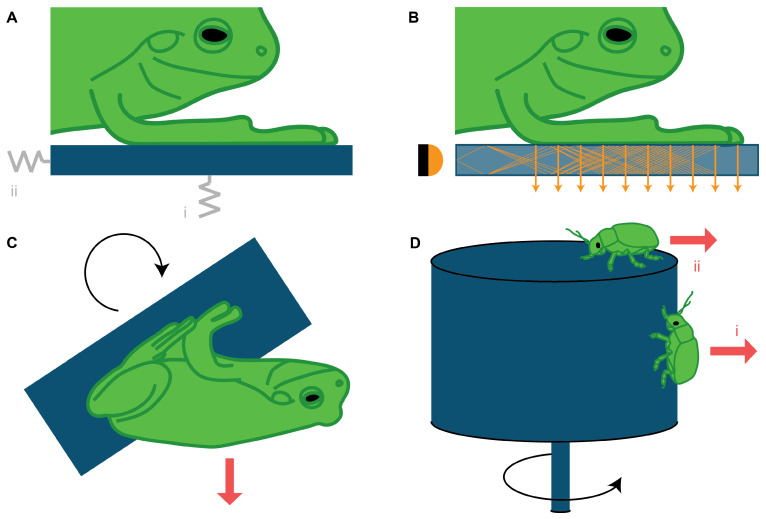
**Measuring attachment forces with global forcing.** (**A**) Force platform interacting with the limb of an unconstrained animal. The springs (in grey) represent capabilities of measuring adhesion (i; normal to substrate) and friction (ii; parallel to substrate). (**B**) Optic sensor based on frustrated total internal reflection (FTIR) to measure the contact area of adhesive pads. The yellow lines represent light reflected inside the transparent substrate, while the yellow arrows represent light that escapes the substrate when it is reflected by the adhesive pads in contact. (**C**) Rotation platform where the animal is gradually rotated around a horizontal axis until the component of gravitational force (red arrow) normal to the substrate exceeds the animal’s adhesive capability. (**D**) Centrifuge system where the rotational velocity gradually increases until the centrifugal force (red arrows) exceeds the animal’s (i) adhesive or (ii) frictional capabilities.

**Figure 3 biomimetics-07-00134-f003:**
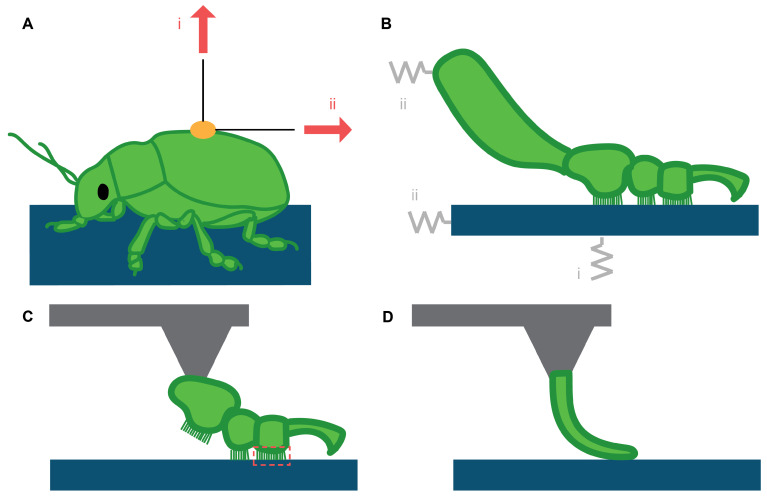
**Measuring attachment forces with local forcing.** (**A**) Tethered experiments where a wire is attached to an animal to measure (i) adhesion or (ii) friction forces. (**B**) Measurements on a limb using force transducers (FTs) to measure (i) adhesion and (ii) friction (or shear) forces. Typically, the shear force is controlled and adhesion measured [8,65]. (**C**,**D**) Atomic force microscopy (AFM) used to measure adhesion of a (**C**) limb and (**D**) its sub-structure, e.g., a seta. Typically, the limb or sub-structure (green) is attached to the AFM probe (grey) and then brought into contact with a substrate (blue) [36].

**Figure 4 biomimetics-07-00134-f004:**
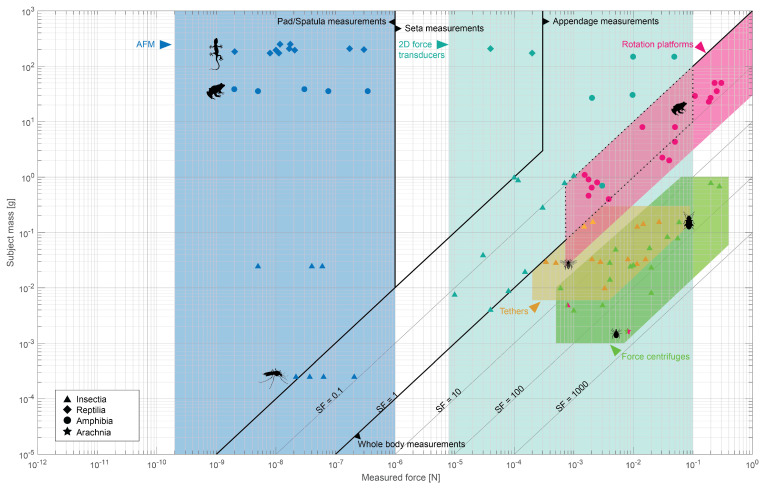
**Ranges of common adhesion and friction force measurement techniques: AFM (blue), 2D (biaxial) force transducers (turquoise), force centrifuges (green), rotation platforms (pink) and tethers (orange).** Data points indicate animal mass and measured force per study, with the symbols denoting taxonomic class. Diagonal lines indicate constant safety factor (SF) lines. Thick black lines denote boundaries between measurements on sub-structures (pad/spatula and setae, respectively), limbs, and whole animals. The area in between the dotted lines shows an overlap of the ranges of limb and whole body measurements. Reviewed studies investigated animals that range across six orders of magnitude in mass, and reported forces that range across nine orders of magnitude. Two studies within the ’force centrifuges’ region are shown with two colors, indicating the study made use of two methods, namely rotation platforms and force centrifuges.

**Table 1 biomimetics-07-00134-t001:** **Summary of bioadhesion measurement methods**.

Level ^1^	Forcing ^2^	Method	Configuration	Subject Class	Dependent Variables	Independent Variables	Measurable Range	Study
Wh	Gl	3D force platforms	Single platform	Geckos	Reaction force	Walking direction	-	[42,43]
Wh	Gl			Tree frogs	Reaction force	Walking direction		[44]
Li	Gl			Insects	Reaction force	-		[45,46]
Wh	Gl		Force Measurement Array (FMA)	Geckos	Reaction force	Surface roughness		[43]
Wh	Gl			Tree frogs	Reaction force	Surface roughness, platform angle		[15,47,48]
Wh	Gl	Photo-elastic gelatin	-	Insects	Reaction force	-	-	[49]
Wh	Gl	Frustrated total internal reflection (FTIR)	-	Insects	Contact area	Load		[50]
Wh	Gl		-	Tree frogs	Contact area	Substrate curvature		[47,51]
Wh	Gl		Rotation platform	Tree frogs	Contact area	Surface roughness		[9,10]
Li	Gl	Optic tactile	-	Geckos	Normal stress	Load angle		[52]
Wh	Gl	Rotation platforms	-	Arachnids	Adhesion %	Surface roughness	{0.7 mN, –} SF = {0.1, 7.0}	[37]
Wh	Gl			Insects	Adhesion %	Surface type, roughness, and structure		[53,54]
Wh	Gl			Tree frogs	Adhesion and shear force	Surface roughness		[10,18,55]
Wh	Gl	Force centrifuges	Adhesion	Insects	Adhesion force	Angular velocity, subject orientation	{500 µN, 500 mN}	[40,41,56,57,58,59,60]
Wh	Gl		Friction	Insects	Dynamic friction force	Surface chemistry and roughness, angular velocity		[37,60,61,62,63,64]
Wh	Lo	Tethered studies	Adhesion	Geckos	Adhesion force	Load	{200 µN, 10 mN}	[65]
Wh	Lo		Friction	Insects	Static friction force	Surface chemistry and roughness		[30,31,32,54,57,66,67,68,69,70]
Li	Lo	1D (uniaxial) force transducers	Adhesion	Insects	Adhesive force	Preload, retraction speed	{80 µN, 100 mN}	[19,71,72]
Li	Lo		Friction	Geckos	Friction force	Surface curvature and roughness, retraction speed		[73]
Li	Lo	2D (biaxial) force transducers	-	Geckos	Friction force	Surface chemistry, preload		[2,3]
Li	Lo			Insects	Friction force	Surface roughness, humidity, preload, sliding speed, retraction speed		[41,74,75]
Li	Lo			Tree frogs	Friction force	Surface roughness, preload		[18,47,76]
Li	Lo	Multiaxial force transducers	3-axis	Geckos	Friction force	Drag direction		[77]
Li	Lo		6-axis	Geckos	Friction force	Substrate roughness		[78]
Su	Lo	Atomic force miscroscopy (AFM)	-	Geckos	Adhesion force	Surface roughnes and chemistry, humidity, preload	{200 pN, 1 µN}	[3,79,80,81]
Su	Lo			Insects	Adhesion force	Surface roughness, humidity		[36]
Li	Lo			Insects	Adhesion force	Buffer presence		[82]

^1^ Wh = Whole animal, Li = Limb, Su = Sub-structure; ^2^ Gl = Global forcing, Lo = Local forcing.

## Data Availability

Not applicable. All data is provided in the text, figures, and table.

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
