# Peer review of "Studying Stickiness: Methods, Trade-Offs, and Perspectives in Measuring Reversible Biological Adhesion and Friction"

_biomimetics, 2022, doi:10.3390/biomimetics7030134_

Round 1

Reviewer 1 Report

The manuscript provides a systematic review of existing techniques for measuring biological adhesion and friction, as well as the challenges that remain in the field, which will offer guidance to scientists that are new to the field of bioadhesion. Overall, the manuscript is well-organized and written but there are some aspects that need to be revised before recommendation for publication.

1.      In section 2.3, the friction measuring of a limb or toe can be greatly affected by the adhesion and the measuring process, since both the force balance and moment balance need to be established for contact. During friction measuring, an adhesion stress could form at the front of contact area, while an extrusion stress could appear at the back of contact area, which bring much difficulty for accurate characterization of friction force and mechanisms establishment of strong attachment. This could be presented in the manuscript.

2.      A new micro-characterizing method for interfacial force has been introduced by observing the movement of thin film interference fringes induced by interfacial deformations, which could in-situ present the movement of liquid film and the corresponding change of force strength, as show in L. W. Zhang, H. W. Chen, et al. Adv. Sci. 2020, 7, 2001125.  This part should be added to the manuscript.

Author Response

We thank the reviewer for his/her useful feedback. We have implemented the suggested edits in the manuscript. Please see the attachment for our point-by-point response.

Reviewer 2 Report

The article presents a broad overview of existing methods to study attachment of terrestrial animals.

The authors reasonably note that deciding on a method for a new study, one should consider a number of practical questions about the parameters need to be measured, including the magnitudes of the force, contact area, stress to be measured and the method suitable for the animal of interest.

The particular manuscript is devoted to the adhesive forces, which correlate strongly with contact area. It is stressed, that every accurate measuring of the related parameters poses a number of challenges. For example, to measure maximum adhesion and friction performance, one needs to detach the animal from a substrate through external forcing, which can be applied globally, as a field, like gravitational or centrifugal forces, or locally by pulling on parts of the animal. The forces can be applied to the entire animal or one of its organs. Behavior of live animals also needs to be considered, because its changes due to rectrictions may lead to errors in interpretation. In this context, there are at least two important points: how the method provides the freedom to vary experimental conditions and how it limits the behavior of an animal.

To help with the choice between alternative methods available the study the authors present relevant considerations when selecting a method, consider some of the limitations of the most prevalent methods with respect to scale of species and their body parts. The authors review the most-used force measurement methods considering whole animals, isolated limbs and their sub-structures and whether the animals experience global or local forcing. Schematically different methods are reproduced in the figures 1a, 1b and 1c of manuscript, respectively. The summary of all the methods is accumulated in the Table 1. As for me, it is the mostly important impact of the paper under consideration.

The authors conclude this review with a novel perspective on force measurement methods focusing on force magnitudes and how they are generated by and/or applied to the animal. Besides an outlook of probable future developments and the general implications of animal adhesion studies is done.

The article provides guidance for scientists that are new to the field of bioadhesion, and presents key challenges in measurement methodology that need to be overcome to advance the field. It is well written, presents necessary illustrative pictures and contains sufficiently long list of the references related to all the techniques reproduced in the main text.

I recommend to publish it in the present form.

Author Response

We thanks the reviewer for his/her recommendation to publish our paper.